# Comprehensive Analysis Identifies Ameloblastin-Related Competitive Endogenous RNA as a Prognostic Biomarker for Testicular Germ Cell Tumour

**DOI:** 10.3390/cancers14081870

**Published:** 2022-04-07

**Authors:** Tianxiang Geng, Catherine Anne Heyward, Xi Chen, Mengxue Zheng, Yang Yang, Janne Elin Reseland

**Affiliations:** 1Department of Biomaterials, Faculty of Dentistry, University of Oslo, 0455 Oslo, Norway; tianxiang.geng@odont.uio.no (T.G.); yang.yang@odont.uio.no (Y.Y.); 2Oral Research Laboratory, Faculty of Dentistry, University of Oslo, 0455 Oslo, Norway; c.a.heyward@odont.uio.no; 3Department of Medicine 3, Uni-Klinikum Erlangen, Ulmenweg 18, 91054 Erlangen, Bavaria, Germany; xi.chen@extern.uk-erlangen.de; 4Laboratory of Reproductive Biology, Faculty of Health and Medical Sciences, University of Copenhagen, 2100 Copenhagen, Denmark; mengxue.zheng@regionh.dk

**Keywords:** testicular germ cell tumour, ameloblastin, ceRNA, comprehensive analysis

## Abstract

**Simple Summary:**

Testicular germ cell tumour is a common tumour in young males, and although it is one of the most curable cancers, many patients still experience recurrence after the chemotherapy. Tumour recurrence is not detected with high sensitivity by established blood tumour markers. Ameloblastin is identified as an extracellular matrix protein and has shown to be associated with tumour progression. We validated ameloblastin’s expression in testicular tissue, and used comprehensive bioinformatics analysis of 156 patients with testicular germ cell tumour to show that the level of ameloblastin was associated with the time of tumour recurrence after the first cure. In the analysis of ameloblastin differential genes in the tumour, a ceRNA (competing endogenous RNA) regulatory network associated with tumour diagnosis and an independent prognostic factor for the tumour, PELATON (Plaque Enriched LncRNA In Atherosclerotic And Inflammatory Bowel Macrophage Regulation), were identified, which could provide evidence for prediction of tumour prognosis.

**Abstract:**

Testicular Germ Cell Tumour (TGCT) is one of the most common tumours in young men. Increasing evidence shows that the extracellular matrix has a key role in the prognosis and metastasis of various human cancers. This study analysed the relationship between the matrix protein ameloblastin (AMBN) and potential biological markers associated with TGCT diagnosis and prognosis. The relationship between AMBN and TGCT prognosis was determined by bioinformatic analysis using the expression profiles of three RNAs (long non-coding RNAs (lncRNAs), microRNAs (miRNAs) and mRNAs) from The Cancer Genome Atlas (TCGA) database, and available clinical information of the corresponding patients. Prediction and validation of competitive endogenous RNA (ceRNA) regulatory networks related to AMBN was performed. AMBN and its associated ceRNA regulatory network were found to be related to the recurrence of TGCT, and LINC02701 may be used as a diagnostic factor in TGCT. Furthermore, we identified PELATON (Plaque Enriched LncRNA In Atherosclerotic And Inflammatory Bowel Macrophage Regulation) as an independent prognostic factor for TGCT progression-free interval.

## 1. Introduction

Testicular Germ Cell Tumour (TGCT) is the most common malignancy in men between 20 and 40 years of age [1]. It is estimated that the number of new European TGCTs will increase by 24% per year by 2025 compared to 2005 [2]. Depending on the tissue in which the tumour occurs, TGCTs are broadly divided into two histological groups: seminoma and non-seminoma [3]. Due to the efficacy of cisplatin chemotherapy, TGCT has become one of the most curable solid cancers [4], and the 5-year survival rate after chemotherapy for patients with testicular cancer is as high as 95% [5]. However, recurrence after the end of chemotherapy still occurs in 20% of TGCT patients, and patients who develop cancer recurrence have a worse prognosis [6]. Blood tumour markers, such as the α-fetoprotein (AFP) and β-human chorionic gonadotropin (β-HCG), have limited sensitivity for detecting disease recurrence. Lesions in the liver or cannabis smoking can cause increased AFP levels and reduce the sensitivity of the diagnosis of TGCT [3]. Even in patients with seminoma, β-HCG is less than 20% effective in diagnosing the disease [7]. TGCT after recurrence will metastasise to other organs such as retroperitoneal lymph nodes [7]. The determination of potential diagnostic and/or prognostic markers is essential for improving the prognosis of patients with TGCT.

The extracellular matrix (ECM) provides mechanical support for cell growth and actively participates in signal transduction processes, but also has a bidirectional influence on tumour recurrence and metastasis of tumour cells [8,9]. It has been shown that the ECM has a powerful influence on tumour invasion and prognosis [10,11]. ECM remodelling is also closely linked to tumour progression [12]. Ameloblastin (AMBN) was discovered as an enamel matrix protein [13] and proven to play an essential role in the development of tooth enamel, cranial bone [14] and long bone [15]. Zhang et al. [16] demonstrated that AMBN can affect cell adhesion, and postulated that these effects are mediated by the well-conserved heparin-binding domains, CD63 interaction domains and calcium binding sites of AMBN. The function of the heparin structural domain of AMBN has recently been investigated in the proliferation of enucleated cell tumours [17]. Xu et al. [18] analysis demonstrated that AMBN might be a potential independent prognostic factor for prostate cancer.

Long non-coding RNA (lncRNA) is a subtype of ncRNA greater than 200 nt in length that has no or only limited participation in protein-coding processes [19]. lncRNAs have several ways of performing their biological functions: it can influence the function of proteins by interacting with one or more protein partners, or it can influence the expression of transcription factors by binding to specific transcription factors [20,21]. lncRNAs have been shown to be extensively involved in developing different types of cancer [22]. In TGCT, specific lncRNAs are associated with cancer recurrence, metastasis and long-term prognosis [23,24]. Therefore, it is of high clinical importance to explore the lncRNA biomarkers associated with the diagnosis and prognosis of TGCT.

MicroRNAs (miRNAs) are a group of non-coding RNAs approximately 22 nt in length that perform essential roles in regulating transcription. They can bind to the 3′-untranslated region (3′-UTR) of the target mRNA to inhibit the degradation or translation of genes and play critical regulatory roles in various physiological and pathological processes [25]. miRNAs regulate cellular functions to influence the response to external stimulation such as hypoxia [26], and oxidative stress [27], and are therefore indirectly linked to cancer development. In addition, many miRNAs directly participate in the pathological process of cancer by acting as oncogenes or tumour suppressors by themselves (Oncomirs) [28]. The competitive endogenous RNA (ceRNA) hypothesis suggests that lncRNAs regulate mRNAs primarily through ceRNA regulatory mechanisms [29] and ceRNAs have been found to have a critical role in human cancer [30].

The present study aimed to explore the relationship between AMBN, its related ceRNAs and the prognosis of TGCT.

## 2. Materials and Methods

### 2.1. Data Preparation

Clinical information and raw data (RNA sequencing data profiles) were analysed for the TGCT patients from the TCGA database (https://portal.gdc.cancer.gov/, accessed date 15 September 2021). Patients’ primary diagnoses included: embryonal carcinoma, seminoma, teratoma, mixed germ cell tumour, yolk sac tumour, teratocarcinoma (for more information see Appendix A). TPM RNAseq data from the UCSC XENA Project (https://xenabrowser.net/datapages/, accessed date 15 September 2021) [31], which included the TCGA and GTEx RNA sequencing data, were analysed together to increase the reliability of data analysis. Available mRNA sequencing (mRNA-seq) data were obtained from the TCGA database for 154 TCGA samples and the UCSC XENA database for 165 normal testis samples. The FPKM (fragments per kilobase per million) data was converted to TPM (transcripts per million reads) format.

### 2.2. Immunofluorescence of Testis Tissues from Rat

After isolation from the 12-week-old Sprague-Dawley male rat, the testis was fixed using 4% paraformaldehyde (PFA) for 24 h, then overnight in cryoprotectant solution [32]. The sample was embedded in optimal cutting temperature (OCT) compound (Leica, Buffalo Grove, IL, USA), and frozen in liquid nitrogen before equilibration at −20 °C for sectioning. Three 5 μm-thickness sections of the testis tissues were used. The animal was part of an experiment was approved by the National Animal Research Authority (approval number 25785). Tissues were sectioned into 5 μm-thickness and mounted onto positively charged glass slides, then antigen-retrieved in Tris-EDTA buffer (10 mM Tris base, 1 mM EDTA solution, 0.05% Tween 20, PH 9.0) 60 °C overnight. Tissue sections were permeabilised with 0.1% Triton X-100 for 15 min, washed with PBS and stained with anti-AMBN antibody (1:200, Affinity Biosciences, Cincinnati, OH, USA) labelled with Goat anti-Rabbit IgG (H + L) Cross-Adsorbed Secondary Antibody Alexa Fluor 488 (1:500, Thermo Fisher Scientific, Carlsbad, CA, USA), and anti-Vimentin antibody (1:200, Sigma-Aldrich, St. Louis, MI, USA) with Goat anti-mouse IgG (H + L) Cross-Adsorbed Secondary Antibody Alexa Fluor 568 (1:500, Thermo Fisher Scientific, Carlsbad, CA, USA), then stained with Deep Red Anthraquinone 5 (DRAQ5, Invitrogen, Carlsbad, CA, USA) to label cell nuclei. The anti-AMBN antibody and anti-vimentin were diluted in 2% normal goat serum, and the secondary antibody was in 4% normal goat serum. The immunolabelled samples were observed under a confocal fluorescence microscope (Leica TCS SP8, Leica Microsystems CMS GmbH, Mannheim, Germany) with HC PL APO CS2 40×/1.30 oil immersion objective lens, running software LAS X 3.5.6.21594 version. Excitation at 488 nm and 5 nm bandpass filters from 500–590 nm. Identical settings were used to scan a negative control sample that had been processed for imaging without a primary anti-AMBN antibody. Excitation and bandpass emission wavelengths were 552 nm and 580–625 nm (anti-vimentin, and Alexa Fluor 568). DRAQ5 imaging of the same field of view used excitation at 638 nm and a bandpass emission filter at 670–750 nm. Images were prepared for publication using Photoshop CS6 (Adobe Inc., Berkeley, CA, USA)

### 2.3. The RNAseq Data Analysis of AMBN

Statistical analysis of mRNA levels of AMBN in 33 types of cancers, such as adrenocortical carcinoma, glioblastoma multiforme and kidney chromophobe (Appendix A), was performed with the RNAseq data in TPM format processed by the Toil process for TCGA and GTEx, visualised using the ggplot2 (version 3.3.3) package. Due to insufficient sample size, TGCT samples were not grouped by histology. Mutations in AMBN in TGCT patients were analysed using cBioPortal (https://www.cbioportal.org/, accessed date 15 September 2021) [33].

### 2.4. Differential Gene Expression Analysis

Patients were categorised into low or high groups based on the gene expression of AMBN, using the median expression as the cut-off value (low expression group: 0–50%, and high expression group: 50–100%). Differential expression analysis was performed using the R package: DESeq2 (version 1.26.0) [34]. A threshold of lncRNA was used for adj *p* < 0.05 and |log fold change (FC)| > 0.5, and *p* < 0.05 and |logFC| > 0.3 as miRNA thresholds, and adj *p* < 0.05 and |logFC| > 0.5 as mRNA thresholds to obtain DERNAs (including different expression lncRNAs, different expression miRNAs and different expression mRNAs). Correlation analysis of DERNAs with AMBN was performed using the stat package (version 3.6.3), and the volcano plot and co-expression heat map were visualised using ggplot2 (version 3.3.3).

### 2.5. Survival Analysis and Construction of Gene-Specific Prognosis Models for TGCT

The public patient data material used here contains copy number data complemented with relevant clinical information. Statistical analysis of survival data was performed using the R package SURVIVAL (version 3.2-10) to analyse Kaplan-Meier survival curves between the high level and low-level AMBN groups. The visualisation was performed using the survminer package (version 0.4.9). Hazard ratios (HR) and 95% confidence intervals (CI) were analysed with Cox proportional hazards regression models to identify factors associated with the study’s primary endpoint. Receiver operating characteristic (ROC) curves of different factors were analysed using the pROC package (version 1.17.0.1) to compare the predictive accuracy and risk scores of the genes of interest. ROC curves were visualised using the ggplot2 package (version 3.3.3).

### 2.6. Immune Infiltrate Levels Related to AMBN

The R GSVA package (version 1.34.0) was used for the expression analysis of the RNA sequencing data of the TGCT patients for the 24 types of tumour-infiltrating cells and the expression of AMBN [35]. The identified tumour-infiltrating cells included: activated dendritic cells (aDC); B cells; CD8 T cells; Cytotoxic cells; DC; Eosinophils; iDC (immature DC); Macrophages; Mast cells; Neutrophils; NK CD56bright cells; NK CD56dim cells; NK cells; pDC (Plasmacytoid DC); T cells; T helper cells; Tcm (T central memory); Tem (T effector memory); Tfh (T follicular helper); Tgd (T γ δ); Th1 cells; Th17 cells; Th2 cells; regulatory T cell (Treg). The markers of the cells from Bindea et al. [36]. In addition, the correlation between immune cells infiltrating the tumour tissue and the level of AMBN was analysed.

### 2.7. Functional Enrichment Analysis

Functional enrichment analysis was performed on the factors obtained from the differential analysis using the clusterProfiler package (version 3.14.3) [37]. Gene ontology (GO) enrichments (including biological process (BP), cellular component (CC) and molecular function (MF)), and Kyoto Encyclopedia of Genes and Genomes (KEGG) pathway enrichments were obtained and ID transformations were performed using the org.Hs.eg.db package (version 3.10.0). False discovery rate (FDR) < 0.25 and *p*.adjust < 0.05 conditions were met for significant enrichment. Gene Set Enrichment Analysis (GSEA) analysis [38] was performed using all seven MSigDB Collections (h.all.v7.2.symbols.gmt (Hallmarks), c2.all.v7.2.symbols.gmt (Curated), c3.all.v7.2.symbols.gmt (Motif), c4.all.v7.2.symbols.gmt (Computational), c5.all.v7.2.symbols.gmt (Gene ontology), c6.all.v7.2.symbols.gmt (Oncogenic signatures), c7.all.v7.2.symbols.gmt (Immunologic signatures)).

### 2.8. Establishment of the ceRNA Network Related to AMBN in TGCT

Combinatorial prediction is employed in target RNA detection. miRNA and mRNA that is explored to target the indicator gene by both algorithms is considered the target RNA. The potential candidate miRNAs for DElncRNAs were identified by LncBase Predicted v.2 (https://carolina.imis.athena-innovation.gr/diana_tools/web/index.php?r=lncbasev2%2Findex-predicted, accessed date 15 September 2021) [39]. The forecasting of mRNAs specific to differential express RNA (DEmiRNAs) was done by miRWalk (http://mirwalk.umm.uni-heidelberg.de/, accessed date 15 September 2021) [40]. The prediction results are taken to intersect with those of TargetScan (http://www.targetscan.org/vert_80/, accessed date 15 September 2021) [41]. Cellular localisation of lncRNAs was optimised using lncLocator (http://www.csbio.sjtu.edu.cn/bioinf/lncLocator/#, accessed date 15 September 2021) [42]. R package miRanda (v3.3a) [43] detects base pairing of lncRNA-miRNA and miRNA-mRNA. The Cytoscape plugin cytoHubba was used to identify hub networks, and the generated networks were visualised by Cytoscape software (https://www.cytoscape.org/, accessed date 15 September 2021). The lncRNA and mRNA base sequence information was obtained from NCBI (https://www.ncbi.nlm.nih.gov/nuccore, accessed date 15 September 2021), and the miRNA base sequence information was from miRBase (https://www.mirbase.org/, accessed date 15 September 2021) [44].

### 2.9. Methylation and Expression Analysis of GFAP

The human disease methylation database UALCAN (http://ualcan.path.uab.edu/, accessed date 15 September 2021) [45] was used to evaluate the methylation levels of glial fibrillary acidic protein (GFAP) between TGCT and normal human testis tissues, and among different TGCT pathological stages. Simultaneously, MEXPRESS (https://mexpress.be, accessed date 15 September 2021) [46] was used to analyse the relationship between gene expression of GFAP and its DNA methylation status.

### 2.10. Statistical Analysis

All data were analysed using SPSS 28.0 software (SPSS, Chicago, IL, USA). The Mann-Whitney U test (Wilcoxon rank-sum test) and independent *t*-test were used to calculate the differences between the data of normal human testis tissue vs. the cancer tissue, and between the data of high AMBN group vs. low AMBN group in the TGCT. One-way analysis of variance (ANOVA) with Kruskal-Wallis test and chi-square test were used to assess between-group differences (normal vs. cancer, and high AMBN vs. low AMBN). Non-parametric correlation tests (Spearman) were used for correlation analysis. Univariate Cox regression analysis was performed to analyse the relationship between candidate genes and progression-free interval (PFI). *p* < 0.05 indicated that the difference was statistically significant (*, *p* < 0.05; **, *p* < 0.01; ***, *p* < 0.001).

## 3. Results

### 3.1. Down-Regulation of AMBN Expression and Clinical Value in TGCT

The TCGA and GTEx databases were used to investigate the expression of AMBN mRNA in various normal and cancers tissues. AMBN mRNA expression was only found to be present in kidney chromophobe (KICH), uterine carcinosarcoma (UCS) tissues and TGCT, as well as in normal testicular tissue (Figure 1A and Appendix A), and not transcribed in the other 29 cancer tissues and their corresponding normal tissues analysed. In the current study, TGCT was not analysed according to the classification of seminoma and non-seminoma. Due to sample size limitations, TGCT was analysed only in groups according to different clinical parameters. Immunohistochemical staining of rat tissues confirmed that the AMBN protein was present in normal testicular tissue (Figure 1B, the corresponding negative control image in Appendix A). In the rat testis, positive staining for AMBN protein was present beside the nucleus of the cells in the seminiferous tubules. The mRNA expression of AMBN was found to be significantly down-regulated in TGCT compared to normal tissue (Figure 1A).

We investigated the relationship between the expression level of AMBN mRNA and several clinical factors/results in TGCT patients to identify the clinical significance of AMBN in this cancer. According to the ROC survival curve, as shown in Figure 1C, the relationship between mRNA expression of AMBN and a specific diagnosis of TGCT is low (AUC 0.675, Cl: 0.613–0.736). In addition, analysis of AMBN mRNA levels for different clinical stages suggested that patients with intermediate to advanced TGCT had significantly lower levels of AMBN in their cancer tissues than patients with earlier stages (Figure 1D, Table 1). Although there was a significant difference in the number of cancer stages between the high and low AMBN groups, the overlap of AMBN level between patients in the early-stage and late stages precludes the use of AMBN levels to identify the stage of cancer. The overlap of AMBN level between patients in the early-stage and late stages precludes the use of AMBN levels to identify the stage of cancer. However, in the survival analysis of progress free interval (PFI) in all TGCT patients, we found that low levels of AMBN were associated with longer intervals between tumour recurrences after the initial cure (Figure 1E). AMBN levels were not associated with overall survival. In further prognostic analyses targeting different subgroups of clinical information classification, patients who developed lymphatic invasion metastases and had low levels of AMBN in tumour tissue were found to have longer cancer recurrence intervals (Figure 1F). Interestingly, in TGCT patients without concomitant lymphatic invasion, mRNA levels of AMBN did not correlate with the time to cancer recurrence after the first treatment (Figure 1G). However, this phenomenon was not found in other clinical factors/results subgroups. To identify mutations in AMBN in TGCT patients, we analysed the genome and copy number of AMBN. The deletion mutations of AMBN gene in the TCGA TGCT dataset were shown by OncoPrint plot (Figure 1H).

### 3.2. Analysis of Differentially Expressed Genes (DEGs)

Based on the observed differences in PFI in tumour tissues with respect to AMBN mRNA level, a natural follow-up was to investigate its role and putative effect. The TGCT patients were divided into low- and high-expression groups based on the median AMBN values, and differentially expressed (DE) lncRNAs, DE miRNAs, and DE mRNAs in both groups were identified. In total 483 DE lncRNAs (250 up-regulated, 133 down-regulated), 38 DE miRNAs (27 up-regulated, 11 down-regulated), and 554 DE mRNAs (432 up-regulated, 122 down-regulated) were screened. The volcano plot visually shows the distribution of DE lncRNAs, DE miRNAs and DE mRNAs (Figure 2A–C). The top 15 FC absolute values were selected by molecular correlation analysis, and a molecular mRNA expression correlation heat map was produced (Figure 2D–F).

The symbols and |logFC| of all DEGs were used to perform GO, KEGG and GESA analyses to explore the functions of these genes. As shown in Figure 2G–J, the functions of differential genes and enriched pathways are focused on steroid hormone metabolism, for example, “C21-steroid hormone metabolic process”, “androgen metabolic process”, “androgen biosynthetic process” “steroid dehydrogenase activity” “ovarian steroidogenesis” and “steroid hormone biosynthesis”. The “steroid hormone biosynthesis” has drawn our attention since serum hCG levels provide an important marker for diagnosing TGCT3.

In the phenotype-related GESA analysis results (Appendix A), the function of DEGs was focused on the Hallmark_Spermatogenesis, Hallmark_Androgen_Response, Hallmark_Xenobiotic_Metabolism, Hallmark_Estrogen_Response_Early, Hallmark_Estrogen_Response_Late, Korkola_Embryonic_Carcinoma_Vs_Seminoma_Dn Korkola, and notably DEGs were negatively associated with a variety of inflammatory cell-related gene sets.

### 3.3. Correlation between Immune Cell Infiltration and AMBN mRNA Expression in TGCT

We evaluated the relationship between AMBN mRNA expression levels and immune cell infiltration in TGCT. First, the single sample GSEA algorithm was applied to analyse the correlation between the level of infiltration of 24 immune cell types and AMBN mRNA expression. The results showed a slight negative correlation between the infiltration of pDC, Neutrophils, Macrophages, iDC, pDC and the level of AMBN (Figure 3A,B). Subsequently, we analysed whether there were differences in the levels of infiltration of 24 immune cell types in the high/low AMBN mRNA expression groups. The results showed that the enrichment scores of 8 immune cell types (DC, iDC, Macrophages, Neutrophils, pDC, Tcm, Th1 cells, Th17 cells) were statistically different in the AMBN low and high expression groups (Figure 3C). Only Tcm infiltration in the cancer tissue of low level patients was lower than the high AMBN group.

### 3.4. Construction of a lncRNA-miRNA-mRNA Triplet Regulatory Network and Its Functional Enrichment

DElncRNAs obtained from previous analyses were used as study subjects, and their target miRNAs were predicted using Tarbase (https://carolina.imis.athena-innovation.gr/diana_tools/web/index.php, accessed date 15 September 2021). Five candidate miRNAs were subsequently included in the following analysis after the resulting predicted miRNAs were intersected with 38 DEmiRNAs. The DElncRNAs unrelated to these five candidate miRNAs were omitted, and the remaining 22 candidate lncRNAs were incorporated into the final regulatory network mapping. The miRWalk and TargetScan databases were used for analysis to predict downstream target mRNAs referencing the five candidate miRNAs. The final 235 candidate mRNAs were incorporated into the tertiary regulatory network’s construction after comparing the predicted mRNAs’ results with the DEmRNAs. The AMBN-related lncRNA-miRNA-mRNA triple regulatory network in TGCT was mapped with Cytoscape (Figure 4A). 22 hub RNAs were screened for hub triple regulatory networks using the Cytoscape plugin cytoHubba, including 5 lncRNAs (LINC02701, PELATON (Plaque Enriched LncRNA In Atherosclerotic And Inflammatory Bowel Macrophage Regulation), PAQR9-AS1, FLJ13224, LINC02026), 5 miRNAs (hsa-miR-5587-5p, hsa-miR-4740-5p, hsa-miR-4689, hsa-miR-5587- 3p, hsa-miR-3153), and 12 mRNAs (LMX1A, ZIC4, PSG1, PSG4, INSL3, INSL4, HSD3B1, MAGEA3, MAGEA6, TM4SF20, PDZK1IP1, GFAP) (Figure 4B). Enrichment analysis of the 22 RNA-associated functions (including GO and KEGG) was performed to explore this regulatory network’s function. The results showed that mRNAs involved in functions such as ovarian steroidogenesis, steroid hormone biosynthesis, cortisol synthesis and secretion, aldosterone synthesis and secretion, were particularly abundant in this regulatory network (Figure 4C).

### 3.5. Validation of the ceRNA Network Model

Expression differential analysis between normal and tumour tissues was used to verify the expression levels of lncRNAs and mRNAs of the hub triple regulatory network. The results showed that, except for HSD3B1, 16 lncRNAs and mRNAs were significantly differentially expressed in normal and tumour tissues (Figure 5A,B). The RNAs with significantly higher expression levels in cancer tissues than in normal tissues include PELATON, INSL4, PSG1, PSG4. Significantly lower RNAs than normal tissue were: LINC02701, PAQR9-AS1, FLJ13224, LINC02026, LMX1A, ZIC4, INSL3, MAGEA3, MAGEA6, TM4SF20, PDZK1IP1, and GFAP. Although the expression level of HSD3B1 in cancer tissues was higher than that in normal tissues, the difference was not statistically significant.

Furthermore, considering that the cellular localisation of lncRNAs determines its underlying mechanism, we analysed the subcellular localisation of these five DElncRNAs by performing lncLocator. As shown in the figure (Figure 5C), LINC02701, FLJ13224 and LINC02026 are mainly located in the cytoplasm, PAQR9-AS1 is mainly located in the nucleus, and PELATON is in the cell membrane. By miRanda, the pairings between DE RNAs were predicted separately. Ultimately, we found that LINC02701 could target GFAP expression via hsa-miR-3153 after screening (Figure 5D,E).

### 3.6. Prognostic Analysis of the ceRNA Network Model

For determining whether these RNAs were associated with the diagnostic and prognostic outcome of TGCT, we analysed and plotted the ROC curves of these three RNAs in TGCT, the KM curves of PFI, and the correlation with AMBN (Figure 6). AUC < 0.7 means low accuracy, 0.7 < AUC < 0.9 for general accuracy and AUC > 0.9 is high accuracy. LINC02701 was shown to give limited diagnostic accuracy with TGCT. Meanwhile, the analysis found that low levels of LINC02701 were associated with a longer time to cancer recurrence after the first cure by TGCT. A positive correlation between the mRNA expression of LINC02701/hsa-miR-3153 and the mRNA expression of AMBN was demonstrated by mRNA expression correlation analysis.

We divided the TGCT patients into two groups using the median GFAP level. In the analysis of clinical prognostic parameters between the two groups, we found that although the number of lymphatic invasions differed significantly, the number of patients with different cancer stages was similar in the two groups (Table 2). However, GFAP was not identified as an independent factor for the occurrence of lymphatic metastasis in TGCT by multifactorial Cox regression analysis (Appendix A). Interestingly, we found that PELATON was an independent prognostic factor for TGCT PFI (Appendix A) and single-gene logistic regression found that PELATON was associated with lymphatic metastasis and tumourigenic location in TGCT (Appendix A).

It was reported that abnormal DNA methylation is strongly associated with oncogenesis [47], so we also analysed the DNA methylation of GFAP in TGCT. Although the mean β value was higher in stage 2.3 than in stage 1, only stage 1 was significantly different from stage 2 (Figure 7A, *p* < 0.05). The β value indicates the level of DNA methylation ranging from 0 (unmethylated) to 1 (fully methylated). Different β value ranges are defined to indicate hypermethylation (β value: 0.7–0.5) or hypo-methylation (β value: 0.3–0.25). Furthermore, 20 methylation sites in the DNA sequence of GFAP were positively correlated with their mRNA expression levels (Figure 7B).

## 4. Discussion

Our analysis revealed the association between AMBN and the prognosis of TGCT that the level of AMBN mRNA within the TGCT tissue was associated with the recurrence interval after the first cure. By analysing sequencing information from the TGCT patients between high and low levels of AMBN, we identified a potential ceRNA tertiary regulatory network: LINC02701–hsa-miR-3153–GFAP. LINC02701 is associated with the recurrence interval after the first cure and GFAP may be a diagnostic factor for TGCT. In addition, we found significant differences in the levels of PELATON between patients with high and low levels of AMBN, which was identified as an independent prognostic factor for TGCT progression-free interval. In the present study, we first explored the changes in AMBN mRNA expression in local tissues. Compared to normal testicular tissue, AMBN levels were decreased in tumour tissue. Furthermore, low AMBN levels could be detected at stage 2.3 in TGCT. Stage T2 and Stage T3 mean that the cancer tissue has spread to the surrounding blood vessels, lymphatic vessels, surrounding soft tissues, or even that the tumour has grown into the spermatic cord [48]. In a study on osteosarcoma, Toshinori et al. [49] found that AMBN inhibited colony formation and migration of osteosarcoma cells through the Src-Stat3 pathway, thereby affecting the severity of osteosarcoma. Moreover Xu et al. [18] demonstrated the feasibility of AMBN as a prognostic biomarker in univariate and multivariate Cox regression analysis and ROC analysis in prostate cancer.

Cancer cell growth is influenced by the infiltration of immune cells into the tumour micro-environment, and multiple immune cell phenotypes have prognostic significance [50]. AMBN has been found to stimulate the expression and secretion of several inflammatory factors in human primary osteoblasts (NHOs) and mesenchymal stem cells (MSCs) [51]. In addition, AMBN up-regulates the inflammatory response of human macrophages, which plays a vital role in innate immunity [52]. CD34+ cells were found to be present in the normal testicular stroma [53], and are precursor cells for a variety of immune cells [54,55]. Tamburstuen et al. [56] found AMBN mRNA expression in CD34+ cells, which may implicate a more direct relationship between AMBN levels and immune infiltration in TGCT tissue. It cannot be ruled out that CD34+ cells were one of the cell types that stained positively for AMBN in the testis, however, the ability to express AMBN may be lost during differentiation to more mature immune cells [54]. and in TGCT, the AMBN levels were found to be negatively correlated with the degree of dendritic cell, macrophage, and neutrophil infiltration by our immune infiltration analysis. The negative correlation of AMBN with the degree of immune cell factors was revealed in the GESA functional clustering analysis, although previous studies have reported that immune infiltration has a minor effect on the overall survival of patients in TGCT [57,58]. If AMBN is expressed during spermatogenesis, a reduction in AMBN may be due to cancer-induced changes in cell differentiation [59], tissue infiltration, or other changes in the cellular micro-environment.

Dendritic cells are associated with better prognosis and lower cancer recurrence in various cancers [60]. Most immune cells, such as DC, had a higher level of infiltration in the cancer tissue of patients in the low-level AMBN group than in the high-level group. The higher level of DC infiltration in the TGCT tissue of the low AMBN group may explain the interesting phenomenon that the relationship between AMBN levels and disease severity is the opposite of the relationship between AMBN levels and PFI.

ceRNA regulatory networks are involved in the occurrence and development of many human cancers, but very few studies have focused on whether ceRNAs can be used as a diagnostic basis for TGCT or to predict TGCT prognosis. In this study, we sought to establish an AMBN-related ceRNA network in TGCT and link it to the diagnosis or prognosis of TGCT. Our enrichment results also show significant differences in the factors related to “ovarian steroidogenesis” between the high and low AMBN mRNA expression groups. It has been demonstrated that increased oestrogen production is associated with reduced spermatogenesis in men with testicular cancer [61], and perhaps AMBN levels may indirectly reflect reduced spermatogenesis in men with testicular cancer after treatment. A valid regulatory pathway, LINC02701–hsa-miR-3153–GFAP, was identified based on subcellular localisation and sequence alignment validation. LINC02701 has not been previously reported to be associated with cancer, but LINC02701 mRNA expression was significantly raised in patients with Parkinson’s disease compared to normal controls [62]. Additionally, LINC02701 was analysed in the recent SARS-CoV-2 study as being associated with the antiviral function of cells and involved in the coordinated expression of signalling pathways in the immune response [63]. The diagnostic role of miRNAs in TGCT has been explored, and miR371 was found to have a sensitive diagnostic specificity in TGCT [64], whereas hsa-miR-3153 appeared to have decreased plasma levels in EGFR mutation NSCLC patients with primary resistance to TKI [65].

GFAP was initially considered to be a glial cell-specific protein and is essential for the normal function of glial cells [66]. Moreover, it is one of the specific diagnostic factors for glioblastoma multiforme [67]. GFAP is expressed in a variety of cell types [68,69], with recent studies demonstrating the expression of GFAP in the testis [70,71]. Our findings support this, and indicate that GFAP levels in TGCT cancer tissues could also be used as a diagnostic factor. The mRNA level of GFAP was found to correlate with the occurrence of lymphatic metastasis in TGCT. Restrepo et al. [72] found raised levels of GFAP promoter methylation in gliomas led to reduced GFAP expression, and that there was frequently loss of GFAP expression with increasing malignancy. We found a similar phenomenon in our analysis of GFAP promoter methylation in TGCT. We found that levels of GFAP promoter methylation were higher in cancer tissue than in normal tissue, and continued to increase as clinical staging progressed. This implies that aberrant methylation of GFAP may be responsible for its prognostic impact on TGCT, and may also explain the lower mRNA levels of GFAP in TGCT tissues than in normal tissues found by our analysis. Gao et al. [73] found that miR-342-5p could regulate mouse neural stem cell proliferation and differentiation by targeting GFAP. We predicted the targeting relationship between hsa-miR-3153 and GFAP using multiple analysis methods. Also coincidentally, the mature miRNA sequence of hsa-miR-3153 could be paired with MSX2 mRNA targeting. MSX2 has been shown to interact with AMBN in vivo, with overexpression of either AMBN or MSX2 affecting each other’s expression levels, and with knockdown of MSX2 the levels of AMBN were also affected [74]. More focused analyses should be performed to investigate the interaction between the two pairs of regulatory pathways; hsa-miR-3153–MSX2 and hsa-miR-3153–GFAP. Due to the limitations of miRNA data in the database this is not feasible in this study. The complex interrelationship between AMBN, ceRNA (LINC02701–hsa-miR–GFAP) and MSX2 may be another reason for the interesting phenomenon that the expression level of AMBN is lower in advanced TGCT than in earlier stages, but high levels of AMBN in cancer tissue predict a shorter time to recurrence after first cure.

PELATON was first detected in inflammatory bowel disease [75]. Hung et al. [76] subsequently verified that it has macrophage and monocyte specificity and can be elevated in levels in unstable atherosclerotic plaques. In recent studies, it was shown to be an iron death suppressor and one of the oncogenes of glioblastoma multiforme (GBM) [77]. Our analysis also demonstrates that PELATON can be an independent prognostic factor for TGCT progression- free interval and is associated with the lymphatic invasion and the first location of TGCT.

Although the ceRNA-based LINC02701/GFAP axis associated with AMBN has been constructed and appears to be a potentially helpful biomarker for diagnosing and predicting PFI, several limitations must also be noted. Firstly, the binding affinity of lncRNAs, miRNAs and mRNAs to each other obtained by analysis of the predictions needs to be experimentally validated. Second, the function and mechanism of the LINC02701/GFAP axis in TGCT need to be further investigated experimentally. In addition, because of sample size limitations in the TCGA database, our analysis did not group TGCT according to non-seminoma and seminoma, but rather directly grouped TGCT patients according to their clinical parameters for comparative analysis. Future work will address more refined tumour classification by cancer cell type.

## 5. Conclusions

In conclusion, we found that AMBN could be a novel predictor of cancer recurrence for TGCT. Furthermore, we established a network of ceRNAs (LINC02701–hsa-miR-3153–GFAP) associated with diagnosing and predicting PFI in TGCT and an independent prognostic factor (PELATON) for PFI in TGCT.

## Figures and Tables

**Figure 1 cancers-14-01870-f001:**
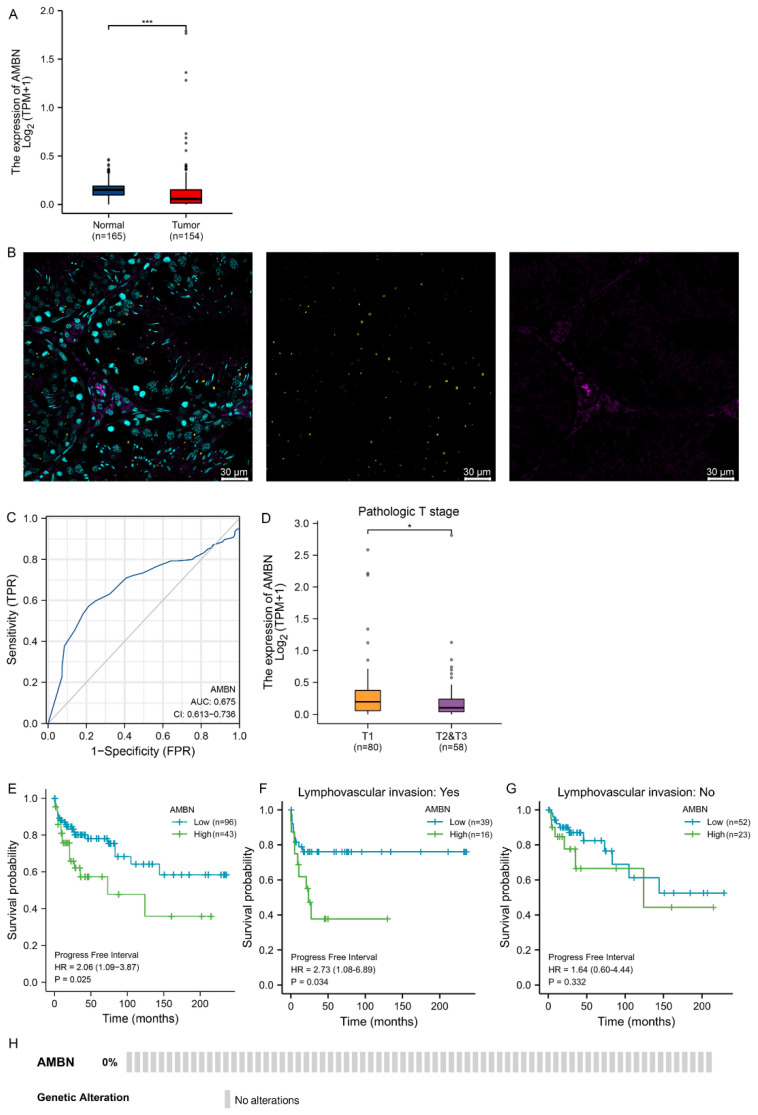
Low mRNA expression of AMBN in TGCT and its clinical implication. (**A**) Distribution of AMBN expression in normal testis and TGCT. (**B**) AMBN expression in rat normal testicular tissue (40×, anti-AMBN (yellow), anti-vimentin (magenta) and DRAQ5 (cyan) to label the nuclei.) (**C**) Receiver operating characteristic (ROC) curve of AMBN in TGCT (**D**) AMBN mRNA expression in different TGCT stages (**E**–**G**) Kaplan-Meier survival curves were used to compare the difference in progress free interval (PFI) between low and high mRNA expression of AMBN under different conditions (**H**) OncoPrint plot showing deletion mutations in AMBN gene in TGCT. (*, *p* < 0.05; ***, *p* < 0.001).

**Figure 2 cancers-14-01870-f002:**
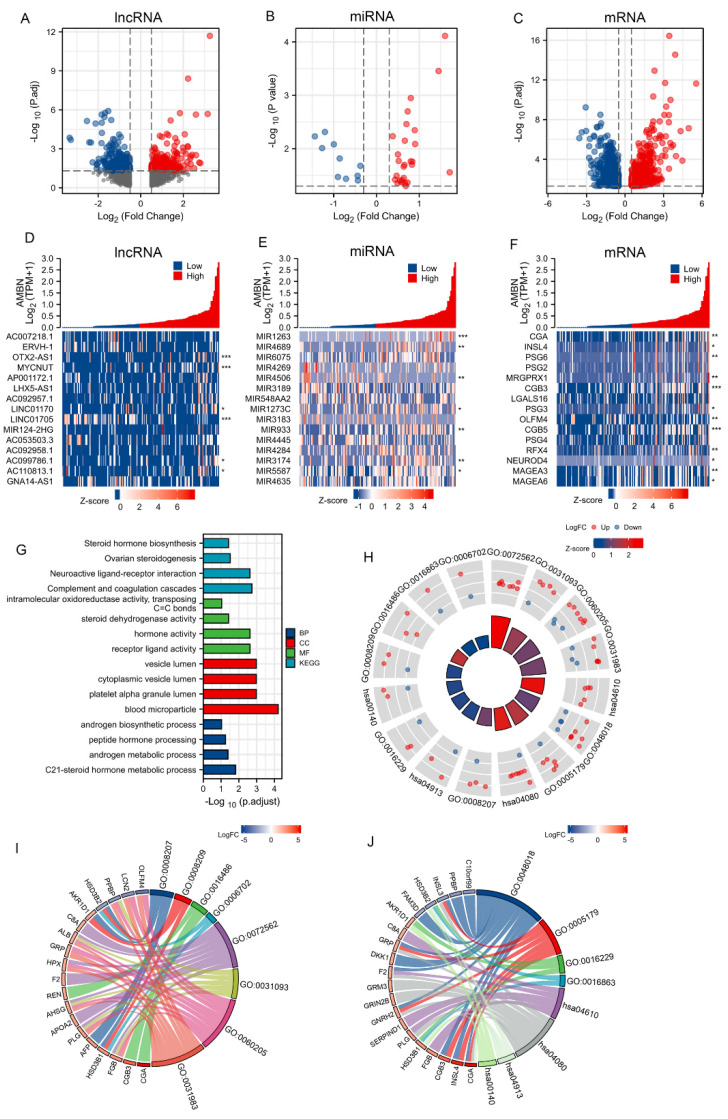
Differential expression gene analysis between AMBN high and AMBN low expression in TGCT samples Volcano plot (**A**–**C**) and heat map (**D**–**F**) for DElncRNAs (adj *p* < 0.05 and |FC| > 0.5), DEmiRNAs (*p* < 0.05 and |logFC| > 0.3) and DEmRNAs (adj *p* < 0.05 and |logFC| > 0.5). and |logFC| > 0.3) and DEmRNAs (adj *p* < 0.05 and |logFC| > 0.5) (z-score indicates the expression level of the factor of interest in the corresponding sample, and * indicates the correlation between the factor of interest and AMBN). (**G**–**J**) GO and KEGG analysis of Differential expression gene (BP for Biological Process, MF for Molecular Function, and CC for Cellular Component, the logFC-score indicates the expression level at the factor of interest, *, *p* < 0.05; **, *p* < 0.01; ***, *p* < 0.001).

**Figure 3 cancers-14-01870-f003:**
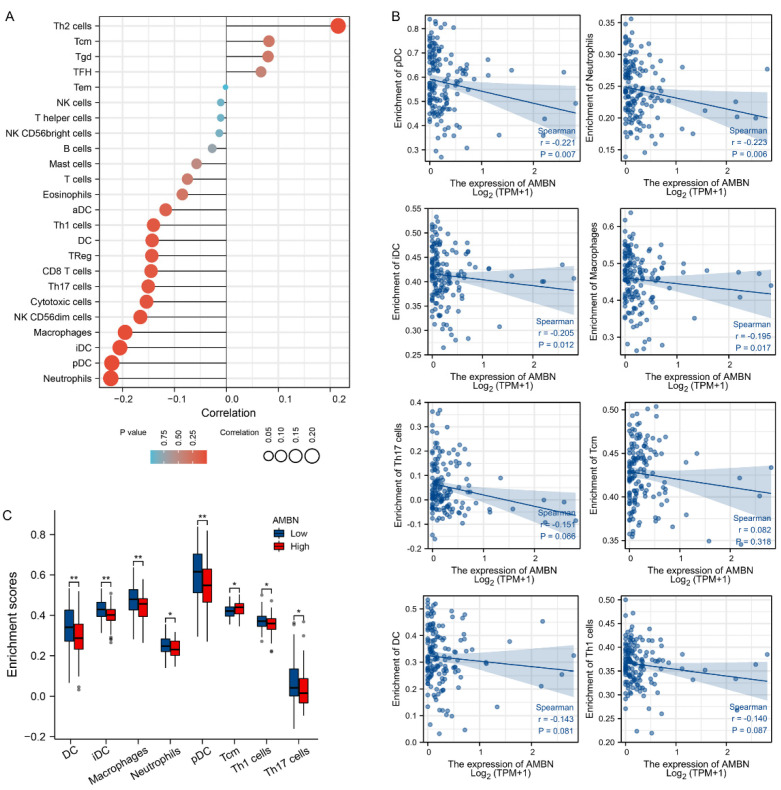
Correlation analysis of AMBN mRNA expression and immune infiltration in TGCT (**A**,**B**) Correlation of AMBN mRNA expression in TGCT with the level of immune cell infiltration. (**C**) Analysis of the differences between AMBN high and AMBN low expression in different TGCT immune infiltrating cells. (*, *p* < 0.05; **, *p* < 0.01).

**Figure 4 cancers-14-01870-f004:**
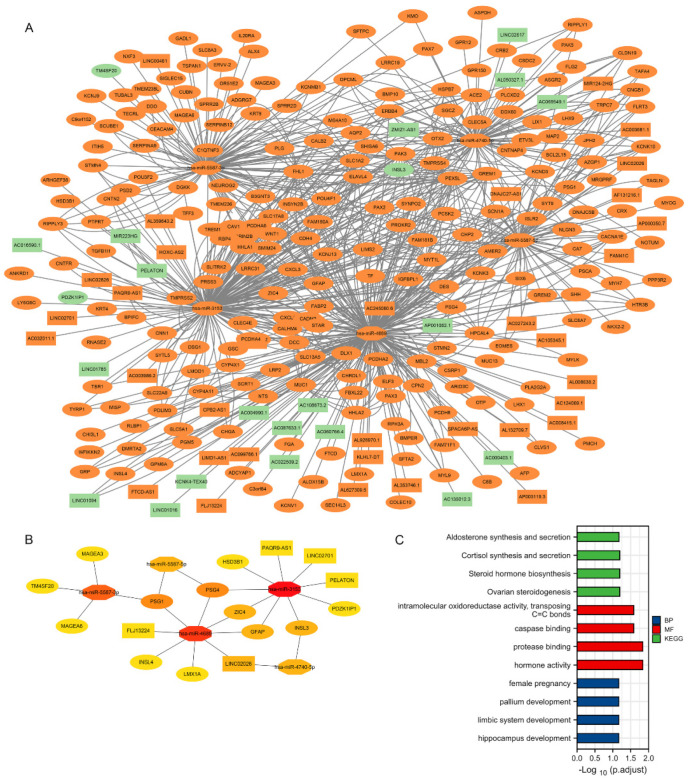
Construction and functional enrichment analysis of the lncRNA-miRNA-mRNA triple regulatory network. (**A**) The triple regulatory network of TGCT. (Red indicates up-regulated mRNA expression, green means decreased mRNA expression; square for lncRNA, polygon as miRNA, circle to mRNA). (**B**) The 22 hub genes in the network (**C**) Functional enrichment analysis of DEmRNAs in the network (GO and KEGG).

**Figure 5 cancers-14-01870-f005:**
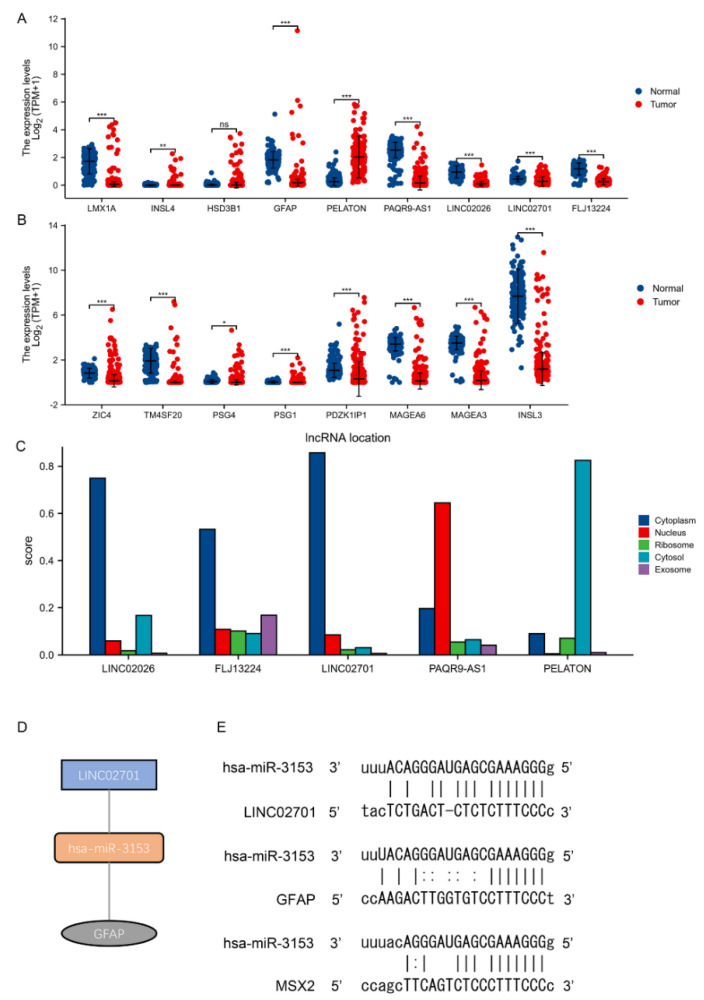
Validation of ceRNA network (**A**,**B**) Expression of hub lncRNAs and mRNAs of the triple regulatory network in the TCGA TGCT dataset (**C**) Predicted cellular localisation of 5 hub-lncRNAs with lncLocator (**D**,**E**) Base pairing between LINC02701-has-miR-3153-GFAP. (*, *p* < 0.05; **, *p* < 0.01; ***, *p* < 0.001).

**Figure 6 cancers-14-01870-f006:**
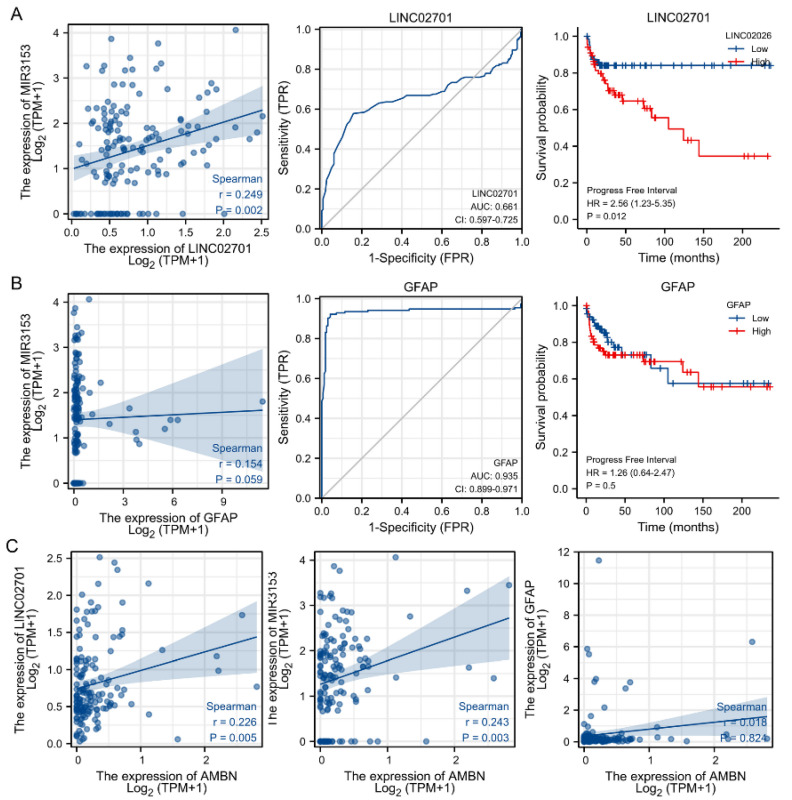
Correlation analysis of ceRNA network (**A**) Correlation analysis of LINC02701 with has-miR3153, ROC and KM curves analysis of LINC02701 with TGCT diagnosis and PFI (**B**) Correlation analysis of GFAP with has-miR3153, ROC and KM curves analysing the significance of GFAP with TGCT diagnosis and PFI (**C**) Correlation analysis between AMBN and LINC02701, has-miR-3153, and GFAP.

**Figure 7 cancers-14-01870-f007:**
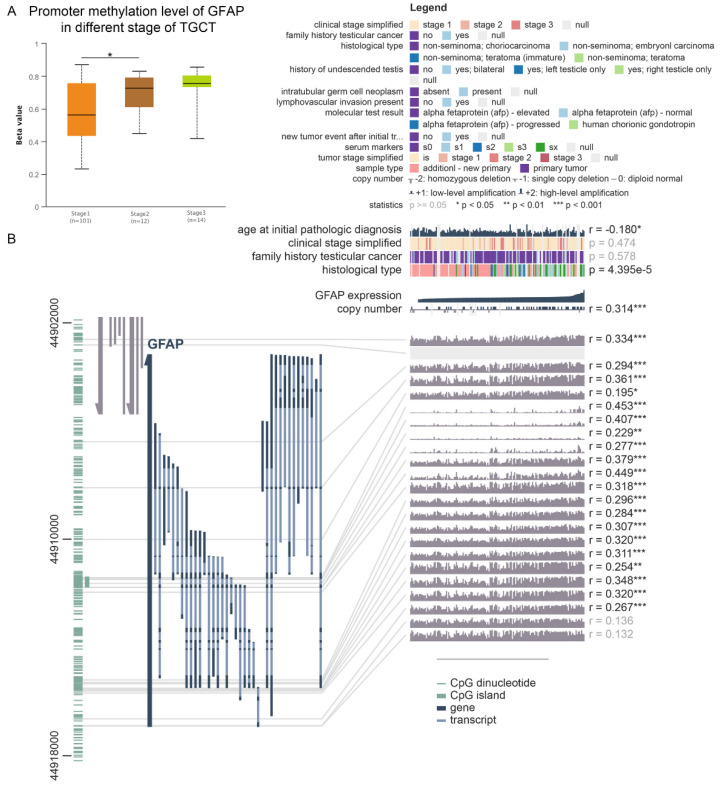
Methylation analysis of GFAP (**A**) Methylation was assessed using DiseaseMeth version 2.0. (**B**) The methylation site of GFAP DNA sequence association with gene expression was visualised using MEXPRESS. *, *p* < 0.05; **, *p* < 0.01; ***, *p* < 0.001.

**Table 1 cancers-14-01870-t001:** Baseline information sheet for AMBN, assessing the difference in the composition ratios of the high and low AMBN mRNA expression subgroups in the different TGCT clinical variables.

Characteristics	Total (N)	Univariate Analysis		Multivariate Analysis	
		Hazard Ratio (95% CI)	*p* Value	Hazard Ratio (95% CI)	*p* Value
AMBN	139	1.590 (0.468–5.400)	0.458		
Pathologic stage	132				
Stage I	106	Reference			
Stage II & Stage III	26	0.210 (0.064–0.691)	**0.010**	11,576,935.157 (1,535,060.618–87,309,534.264)	**<0.001**
Clinical stage	130				
Stage I	98	Reference			
Stage II & Stage III	32	0.202 (0.071–0.576)	**0.003**	0.000 (0.000–0.000)	**<0.001**
Radiation therapy	137				
No	113	Reference			
Yes	24	0.873 (0.386–1.978)	0.745		
Primary therapy outcome	88				
PR & PD	14	Reference			
CR	74	9.339 (1.264–68.994)	**0.029**	155,601,452.260 (0.000–Inf)	0.997
Race	134				
Asian	4	Reference			
Black or African American	6	5.045 (0.520–48.936)	0.163	6,123,473,977.844 (714,602,666.954–52,472,423,195.932)	**<0.001**
White	124	1.192 (0.163–8.728)	0.863	246,966,750.563 (28,820,747.782–2,116,273,191.320)	**<0.001**
Age	139				
<=30	67	Reference			
>30	72	0.697 (0.373–1.301)	0.257		
Serum tumor markers(S)	125				
S0	43	Reference			
S1	41	1.944 (0.782–4.835)	0.153	3.655 (1.527–8.750)	**0.004**
S2	36	2.908 (1.190–7.106)	**0.019**	5.402 (2.230–13.087)	**<0.001**
S3	5	3.993 (0.827–19.283)	0.085	1.000 (1.000–1.000)	
Lymphovascular invasion	135				
No	79	Reference			
Yes	56	1.364 (0.733–2.538)	0.327		
Testicular intratubular germ cell neoplasia	130				
Absent	71	Reference			
Present	59	0.729 (0.383–1.386)	0.335		
History of undescended testis	132				
No	109	Reference			
Yes	23	0.560 (0.219–1.434)	0.227		
Family history of testicular cancer	122				
No	107	Reference			
Yes	15	3.009 (1.407–6.435)	**0.005**	1.637 (0.621–4.314)	0.319
Laterality	134				
Left	74	Reference			
Right	60	1.231 (0.639–2.371)	0.534		
Pathologic T stage	138				
T1	80	Reference			
T2 & T3	58	1.158 (0.621–2.160)	0.645		
Pathologic N stage	64				
N0	51	Reference			
N1 & N2	13	0.108 (0.015–0.793)	**0.029**	1.000 (0.133–7.542)	1.000
Pathologic M stage	124				
M0	120	Reference			
M1	4	0.000 (0.000–Inf)	0.996		
Clinical T stage	118				
T1	68	Reference			
T2 & T3	50	0.832 (0.433–1.596)	0.579		
Clinical N stage	113				
N0	83	Reference			
N1 & N2 & N3	30	0.212 (0.074–0.608)	**0.004**	1.000 (0.133–7.542)	1.000
Clinical M stage	133				
M0	125	Reference			
M1	8	0.624 (0.150–2.600)	0.517		

**Table 2 cancers-14-01870-t002:** Baseline information sheet for GFAP, assessing the difference in the composition ratios of the high and low GFAP mRNA expression subgroups in the different TGCT clinical variables.

Characteristic	Low Expression of GFAP	High Expression of GFAP	*p*
*n*	67	67	
Pathologic stage, *n* (%)			0.303
Stage I	51 (40.2%)	50 (39.4%)	
Stage II	6 (4.7%)	6 (4.7%)	
Stage III	4 (3.1%)	10 (7.9%)	
Clinical stage, *n* (%)			0.370
Stage I	49 (39.2%)	44 (35.2%)	
Stage II	9 (7.2%)	8 (6.4%)	
Stage III	5 (4%)	10 (8%)	
Primary therapy outcome, *n* (%)			0.868
PD	0 (0%)	1 (1.2%)	
SD	0 (0%)	0 (0%)	
PR	5 (6%)	8 (9.6%)	
CR	31 (37.3%)	38 (45.8%)	
Age, *n* (%)			0.057
≤30	26 (19.4%)	38 (28.4%)	
>30	41 (30.6%)	29 (21.6%)	
Serum tumour markers(S), *n* (%)			0.041
S0	28 (23.3%)	15 (12.5%)	
S1	16 (13.3%)	22 (18.3%)	
S2	12 (10%)	22 (18.3%)	
S3	2 (1.7%)	3 (2.5%)	
Lymphovascular invasion, *n* (%)			0.023
No	45 (34.6%)	30 (23.1%)	
Yes	21 (16.2%)	34 (26.2%)	
Testicular intratubular germ cell neoplasia, *n* (%)			0.246
Absent	38 (30.4%)	30 (24%)	
Present	25 (20%)	32 (25.6%)	
History of undescended testis, *n* (%)			0.258
No	47 (37%)	57 (44.9%)	
Yes	14 (11%)	9 (7.1%)	
Family history of testicular cancer, *n* (%)			1.000
No	52 (44.1%)	52 (44.1%)	
Yes	7 (5.9%)	7 (5.9%)	
Laterality, *n* (%)			0.938
Left	35 (27.1%)	37 (28.7%)	
Right	29 (22.5%)	28 (21.7%)	

## Data Availability

Data sharing is not applicable to this article as no new data were created or analyzed in this study.

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
