# Peer review of "Comprehensive Analysis Identifies Ameloblastin-Related Competitive Endogenous RNA as a Prognostic Biomarker for Testicular Germ Cell Tumour"

_cancers, 2022, doi:10.3390/cancers14081870_

Round 1

Reviewer 1 Report

The present study by Geng et al. is based on database analyses and bioinformatics/computing on TGCT data with which the authors aim to identify the relationship between different types of RNA (lncRNAs, miRNAs, mRNAs) and ameloblastin (AMBN) for defining new clinical parameters for TGCT diagnosis and prognosis. In the end, the authors present a regulatory pathway between LINC02701, hsa-miR-3153 and GFAP. Whereas the computing data e.g. the pathway analyses etc is certainly valid as it stands, the biological implications and conclusions drawn from these lack strong clinical evidence and backup. Overall, a clear specification of what TGCT stands for in this study cohort, appears crucial to better understand and interpret the data generated.

  • In this study, AMBN is mentioned as a matrix protein. Usually, it is mentioned in the context of tooth composition. Hence, it would be crucial to understand which cells of the testis may be able to produce ameloblastin, whether it is relevant in the testis as an ECM protein and why its expression is reduced in late stage TGCT (because of loss of the AMBN producing cells?).
  • line 39 ff: The term TGCT (testicular germ cell tumour) as well as the data set used imply that TGCT is ONE clearly defined tumour type. But the term TGCT comprises a group of testicular tumours (GCNIS, seminoma, non-seminoma etc) with very different characteristics when it comes to therapeutic efficiency, prognosis etc. What types of TGCT are combined in the data set analysed? Please add a table/list explaining which types of TGCT are represented in the dataset to better outline the patient material. The authors e.g. group patients based on stage of disease, not type of TGCT. This may make a great difference if different types of TGCT are combined. Also, it would be highly relevant to know whether if you group the patients by TGCT type specifically, can further specifications be made based on which type of cancer has a low vs high AMBN expression etc?
  • The reviewer would appreciate if further specification in results sections/figure legends would be added that clearly specifies the term „AMBN expression“, to clarify if mRNA or protein data is being shown/meant.
  • Fig. 1 the color-coding is a bit confusing. Maybe this could be improved. Only „blue“ and „red“ are being used but for different groups (A+C blue = normal, re = tumour, E blue = T1, red = T2+3, F-H blue = low AMBN, red = high AMBN). This is not very intuitive. Also, it would be great if the legend would be supplemented with n-numbers to provide this important information right where it is needed. E.g. how many patients were stage 1, 2, 3 respectively, or figure 1 F, how many patients were considered AMBN low vs AMBN high.
  • 1 B and lines 211 – 213: The investigators performed immunofluorescence on wildtype rat testicular tissue in order to confirm AMBN (protein) expression. The testis consist of various different types of intratubular and interstitial cells. It is not specified where i.e. in which cell type, which cellular compartment etc staining was detetected. Especially with regards to the later reference to immune cell infiltration, further investigations/stainings/specifications in this regard would be important data. Also, further image labelling, marking seminiferous tubules vs interstitium etc would greatly help the reader to independently interpret the data. Otherwise, since the whole paper deals with human data, it would be stringent if also the protein data shown with IF/IHC would be performed using human tissue samples.
  • line 152: it is unclear to the reviewer where the immune cell data comes from, please specify. In this regard, an enrichment score is being shown in Fig. 3C. Please specify how this score is being created. Especially with „cell count data“ it is important to clarify whether a rather computational/machine based approach such as flow cytometry/FACS was being used or whether data was generated by visual analysis of sections by either an automated technique (ImageJ etc) or by „human eye“. Also, quite unique cell types such as Tcm are being included – it would be important to provide information on how the respective cell types were identified (e.g. which markers were used etc).
  • line 214 – 215: the difference between figure 1A and figure 1 C remains unclear. Where does the data shown in fig. 1 C come from i.e. how does it differ from figure 1 A?
  • Lines 220 – 229: the authors report that in advanced TGCT (stages 2 + 3?), AMBN expression is lower than in stage 1. Also, low AMBN expression was found to be linked to a longer cancer recurrence interval. Does this indicate that survival in low AMBN patients i.e. in late stage cancer patients is better than in stage 1 patients? Please discuss critically and clarify.
  • lines 229 – 230 and figure 1 G + H: it is not intuitive how to interpret these figures. Even in the results section, the explanation is kept very short. Please explain further.
  • line 245: „based on the significant differences in prognosis of AMBN in testicular tumour tissue…“ please clarify how you come to that strong conclusion. Is it considered that any patient that presents with a low AMBN expression, no matter what stage he is in already, will have better chances for life? What about the specific type of TGCT, does this need to be taken into account? Or are there differences in terms of AMBN expression in different types of TGCT or the overall prognosis in different types of TGCT, anyhow? For making such a strong statement, clear arguments should be provided. Please discuss and outline.
  • Line 276 ff: „correlation between immune cell infiltration and AMBN expression in TGCT“. The authors mention that there is a slightly negative correlation between AMBN expression and immune cell infiltration. If the reviewer understands correctly, the low AMBN group comprises the rather late stage TGCT patients. Are immune cell infiltrations more pronounced in late stage tumours per se? Since TGCT and cancers in general are comprised of a whole lot of different cell types, it would be interesting to see, which cell type may be the source of AMBN expression aka why the expression is reduced in late stage TGCT.
  • Figure 2: once again, mainly red and blue are being used. In A – C, the reviewer interprets that blue again means low AMBN group and red = high AMBN group. In D then, it is unclear to which group of samples the heatmap belongs since there doesn´t seem to be a seperate visualisation of heatmaps for the low vs high AMBN group? Please consider rearraging the sub figures or providing more information in the legend.
  • 2 G = Whereas KEGG mentioned on the legend is self-explaining, it is unclear what BP, CC and MF stands for. Please specify somewhere.
  • 2 I + J: Is the z-score relevant for interpretation of these figures? Or does it belong to other figures? This remains unclear to the reader.
  • lines 349 – 350: „LINC02701 was shown to have high diagnostic accuracy with TGCT and may have potential as a basis for TGCT diagnosis.“ The reviewer is curious how the the authors see the practical applicability of LINC02701 as diagnostic tool. What do you assume would be the material to sample LINC02701 from a patient? Also, please discuss, whether the simple fact that a factor, in this case, some RNA is reduced in a certain tumour type makes it a diagnostic target already?
  • Figure 7 A and lines 363 - 366: in this figure, for the first time, a differentiation between seminoma and non-seminoma patients is shown, which would be highly appreciated. In the results section though, the authors report a significant difference in GFAP methylation between normal (testicular) tissue and cancer tissue. This is not coherent. Also, the significance is not indicated in the figure itself. Same for figure 7 B, for the first time, a clear differentiation between results for stage 1, 2 and 3 cancer patients are being shown, including n-numbers. It would be great if such detail would be given in all/more figures.
  • Table 1: it is interesting to the reviewer that the low vs high GFAP expression groups actually show a very similar overall pattern, meaning that it is not the case that e.g. a certain stage of disease is associated with high GFAP expression. To the reviewer, this suggests that there is no correlation between GFAP expression and clinical data, hence GFAP does not appear to be an appropriate marker to seperate patients into one or another group. Please discuss how this works together with your comment on GFAP as a diagnostic marker. Otherwise, such table would be important to include for the correlation between AMBN (the main topic of this paper) and the clinical variables to further validate or present the potentially clinical/biological relevance of low vs high AMBN expression in tumour patients and its practicability of differentiating patients.
  • line 405: „immune cell infiltration in tumour tissue was strongly associated with tumour progression“ – is this comment based on the own observations of the authors? Or is this a citation from someone else´s paper? If so, it should be reworded. Also, line 408, the authors mention that AMBN upregulates the inflammatory response in immune cells. How do you explain the biological relationship then, if in late stage tumours there is more infiltrating immune cells but low AMBN? Is this advantegous or disadvantageous for the whole „organism“? Especially with regards to the fact that the authors mention that the interval between tumour recurrences may still be longer in late stage patients? To the reviewer, the overall biological implications of the findings still remain largely unclear. Also, line 410 ff, the authors cite that AMBN is being expressed in CD34+ cells. In their study though, AMBN expression is low in tumour samples that should have a high immune cell infiltration. This once again comes to the question of which cells are the source of AMBN in the testis?
  • lines 418 ff: „most immune cells, such as DC, had a higher level of infiltration in cancer tissue of patients in the low level AMBN group than in the high-level group.“ Isn´t that because the low level AMBN group comprises the late stage tumour patients which are known to have higher immune cell infiltrations? Please discuss.
  • lines 463 – 466: this is indeed an „interesting phenomenon“ that according to the reviewer, the authors have not discussed in enough detail to convince the reader. Maybe, by further specifying the groups they are comparing (low AMBN vs high AMBN, tumour type, stage of disease etc), things may become more clear.
  • lines 483: „exploring the pathogenesis of TGCT“ is a whole different field of study, especially given the fact that the authors use TGCT samples without classifying the actual underlying tumour types that are believed to harbour a different pathogenesis even if not fully understood yet.

The way patients were assigned to groups does not seem to meet biological criteria.

Reviewer 2 Report

The Authors have provided an interesting paper on ameloblastin-related competitive endogenous RNA in GCT. I would suggest some comments

1) In the introduction section more infornation (brefly) have to be included and not only a line on beta HCG

2) Refrences :

3 has to be updated (March 2021)

6 has to be updated

Some information regarding the cliniclstudies on miRNA 371 cluster have to be added (see Nappi L et al)

3) Please specify the histology of these patients

Round 2

Reviewer 1 Report

  • The authors have addressed most questions adequately and adjusted the wording where needed. Also, further information was provided to ease understanding. The reviewer still believes that it is essential to mention in the introduction that TGCT is not one type of tumour (e.g see Shen et al., doi: 1016/j.celrep.2018.05.039). In the revised version of the manuscript, the different types of tumours combined in the data set TGCT are listed only briefly in materials and methods, without mention of n-numbers, still. To the reviewer, it remains important to add a full table/list explaining which types of TGCT are represented at which numbers. In that sense, also important tumour type characteristics could be added.
  • The authors do not refer to why no human samples were used for immunolocalisation of AMBN but rather rat tissue. Since the whole manuscript deals with human data, a localization of AMBN mRNA/protein within the human testis would be stringent.
  • The authors further claim “…we cannot include more data in this manuscript, but if of interest we can provide the reviewers with preliminary results that indicate the distribution of AMBN in testicular tissue.”. From the reviewer´s point of view, there is certainly uncritical information that can be provided without giving away sensitive or “unproven” data. For example, the image does not include any labelling to differentiate seminiferous tubules from interstitium etc (which in IF images is much harder than in IHC images) and as such doesn´t provide any information other than “there are green dots visible in the tissue section”. This could be easily improved without any danger to the unpublished data.
  • The authors mention that AMBN has been demonstrated to be expressed in CD34+ cells and speculate that a reduction in expression in late stage TGCT may be related to differentiation of CD34+ cells and spermatogonia. Since there is papers on CD34 expression in the testis, it would be supportive data to add in information from these papers giving hints at expression sites of CD34+ cells and potentially consequently AMBN expression in the testis.
  • “Thanks to the reviewer's reminder about the tumour type and analysis grouping. The authors also thought carefully about this issue before conducting the analysis. The grouping based on stage was not the only method used by the authors: the presence or absence of lymphatic invasion and the effectiveness of chemotherapy etc., were also used as a grouping basis. As TGCT is very sensitive to chemotherapy, the number of patients with bad prognoses were limited. In addition, our ongoing research showed the link between AMBN and germ cells, and the separate analysis of TGCT on different tissue sources is one of our research priorities.” The reviewer doesn´t really understand this answer. What do you mean by separate analysis of TGCT on different tissue sources? To the reviewer´s knowledge, analyses on TGCT usually firstly define the different subgroups (at least seminoma vs non-seminoma) before going into any further analysis or comparison of data. If understood correctly, for the present study, the data set “TGCT” was for most experiments either grouped into “high vs low expression of GFAP” or “high vs low expression of AMBN” groups. Somewhere else the authors state. “The reviewer's suggestion of refined groupings is relevant, and it would be useful to study multiple sources of testicular tumours, however this is not the focus of current research, and the data number will not be enough after subgroup.” The authors lastly answer that “data number will not be enough after subgroup”. If this is the explanation why the data set is not grouped based on tumor types but based on different marker expression levels or clinical parameters (such as tumour stage, presence/absence of lymphatic invasion etc), this limitation should be mentioned in the manuscript to give full transparency to the reader and allow him/herself to judge the results accordingly.
  • “Baseline information sheet for GFAP, assessing the difference in the composition ratios of the high and low GFAP mRNA expression subgroups in the different TGCT clinical variables.” The reviewer is curious as to why such table is still not provided for the high and low AMBN mRNA expression subgroups which is a main grouping criteria used in this manuscript. Hence its association with clinical variables would be highly beneficial and important.
  • “The analysis of UALCAN showed that GFAP showed hypomethylated levels in non-seminoma but showed hypermethylated levels in seminoma tissue.” …compared to what? There is no reference value given in the figure or text. The figure only presents the difference between Se and Non-Se. Also, since nowhere else in the manuscript data is differentiated into e.g. seminoma vs non-seminoma patients or e.g. not even the difference between these two tumour types is being explained, this data cannot be put into any reference/perspective.
